# Side Streams of Vegetable Processing and Its Bioactive Compounds Support Microbiota, Intestine Milieu, and Immune System

**DOI:** 10.3390/molecules28114340

**Published:** 2023-05-25

**Authors:** Joanna Fotschki, Anna M. Ogrodowczyk, Barbara Wróblewska, Jerzy Juśkiewicz

**Affiliations:** 1Department of Immunology and Food Microbiology, Division of Food Science, Institute of Animal Reproduction and Food Research, Polish Academy of Sciences, Tuwima 10, 10-748 Olsztyn, Poland; a.ogrodowczyk@pan.olsztyn.pl (A.M.O.); b.wroblewska@pan.olsztyn.pl (B.W.); 2Department of Biological Functions of Food, Division of Food Science, Institute of Animal Reproduction and Food Research, Polish Academy of Sciences, Tuwima 10, 10-748 Olsztyn, Poland

**Keywords:** bio-waste, vegetable waste, vegetable by-products, immune system, intestinal microbiota, vegetable side stream, bioactive compounds from by-products

## Abstract

The industry of vegetable processing generates large amounts of by-products, which often emerge seasonally and are susceptible to microbial degradation. Inadequate management of this biomass results in the loss of valuable compounds that are found in vegetable by-products that can be recovered. Considering the possibility of using waste, scientists are trying to reuse discarded biomass and residues to create a product of higher value than those processed. The by-products from the vegetable industry can provide an added source of fibre, essential oils, proteins, lipids, carbohydrates, and bioactive compounds, such as phenolics. Many of these compounds have bioactive properties, such as antioxidative, antimicrobial, and anti-inflammatory activity, which could be used, especially in the prevention or treatment of lifestyle diseases connected with the intestinal milieu, including dysbiosis and immune-mediated diseases resulting in inflammation. This review summarises the key aspects of the health-promoting value of by-products and their bioactive compounds derived from fresh or processed biomass and extracts. In this paper, the relevance of side streams as a source of beneficial compounds with the potential for promoting health is considered, particularly their impact on the microbiota, immune system, and gut milieu because all of these fields interact closely to affect host nutrition, prevent chronic inflammation, and provide resistance to some pathogens.

## 1. Introduction

Vegetables are an important part of any healthy diet due to the source of essential micro and macro-nutrients, such as vitamins, minerals, and fibres. Due to the seasonality, adaptation to the market, and consumer preferences, most vegetables are processed, generating large amounts of by-products. The global percentage of by-products produced during the food supply chain’s upstream operations reaches 40%, of which 15% is due to agricultural production, 15% generates processing and packaging, and 10% is generated during postharvest handling and storage [1]. Depending on the site of origin and processing technology, vegetable by-products include pomace, pulp, peels, husk, hull, leaves, juices, bagasse, pastes, and kernels [2]. These post-processing side streams still contain many bioactive compounds, such as proteins, lipids, fibre, essential oils, carbohydrates, and phytochemicals (polyphenols and carotenoids) [3]. Many of these compounds perform multiple biological activities, such as anti-inflammatory, antimicrobial, and antioxidative, which are beneficial for human health. This was partly summarised in comprehensive reviews of the health-promoting activity of individual phytochemicals [4,5]. However, the stability of by-products is low due to their high water activity, which is associated with the quick growth of unfavourable microbes and spoiling, but also with auto-oxidation in the case of fat-containing by-products. The majority of vegetable by-products are either used as animal feed and biofertilizers or discarded. Inadequate management of this biomass results in the loss of valuable compounds, which they contain and may be recovered. Economic and legal restrictions are imposed by the costs of by-product drying, technological treatment, storage, shipping, and disposal [6]. Some of the bioactive substances in the side streams can be naturally concentrated during the processing, as in the case of onion husks or carrot pomace. The other case is post-production water, in which the content of bioactive substances may be lower than in the raw material but still significant enough that the utilization of this by-product may be profitable, such as potato or tofu wastewater [7].

In this publication, the authors focused on vegetable by-products that often emerge seasonally and, without any technological treatment, such as drying, are prone to rapid decay. Therefore, we will discuss mainly such phylogenetic groups as plants from the following families: *Solanaceae* (e.g., pepper, tomato, potato, eggplant); *Brassicaceae* (e.g., cabbage, cauliflower, broccoli, Brussels sprout, komatsuna); *Cucurbita* (e.g., pumpkin, cucumber); *Apiaceae* (e.g., carrot); *Amaryllidaceae* (e.g., onion, garlic); *Fabaceae* (e.g., legume, pea, soybean); *Asteraceae* (e.g., chicory, lettuce, artichoke); and *Amaranthaceae* (e.g., sugar beet). In these groups of plants, if the right conditions are not maintained, they spoil very quickly; therefore, their management is a challenge for the food industry, often generating high disposal costs. Bioactive compounds from by-products that could be vital for human and animal health are often unstable when exposed to environmental conditions and industrial processes. 

Due to the deepening economic crisis and the growing demand for nutritious and health-promoting food, the optimal use of post-production waste seems to be essential. There is a need to study vegetable by-products more deeply and find a joint solution to the problems of waste management and resource depletion. In this review, the authors focused mainly on the issue of the usefulness of vegetable-based by-products as a source of biologically active substances with microbiota-regulating, anti-pathogenic, intestinal milieu-regulating, and immunomodulating, as well as cytotoxic and cytoprotective properties. The authors also focused on highlighting the factors that influence the final properties of food industry by-products, such as the isolation of different components, applied stabilization, and the analytical techniques supporting the claimed characteristics. 

## 2. Modulation of the Intestine Milieu and Microbiota

Bacteria, archaea, viruses, and eukaryotic microbes create the human gastrointestinal microbiome, which has tremendous potential to impact our physiology, both in health and disease [8]. The gut microbiota plays an important role in the function of the human gastrointestinal tract through its nutrient, xenobiotic, and drug metabolism, antimicrobial protection, immunomodulation, and maintaining the structure and function of the gastrointestinal cells [9]. The strong relation between balanced gut microbiota and the parameters of the immune system has been reported [10]. Furthermore, the microbiota plays an essential role in the controlling of infection and the ability to promote and calibrate both innate and adaptive immunity [11]. Microbiota dysbiosis causes some illnesses, including those that are regarded as global issues, including inflammatory bowel disease, rheumatoid arthritis, colorectal cancer, obesity, diabetes, mental disorders, and cardiovascular disease [12]. Thus, a microbiota-modulating diet is desirable to provide a good condition of the intestine as well as the immune system. 

### 2.1. The Effect on Intestinal Microbiota

The intestinal microbiome-associated metabolites have a decisive influence on the host’s metabolism and are essential for multiple functions, such as modulation of organ-specific immune responses, through translocation from the intestinal lumen to various organs (e.g., liver, brain or lung) and subsequently, induce tissue-specific local immune responses [13]. To maintain the balance in the gut microbiota, there is a need to support beneficial intestinal bacteria by providing them with suitable substrates, such as dietary fibres, to prevent the growth of harmful bacteria and other pathogens. Many scientific reports indicate that vegetable by-products affect microbiota, suggesting health-promoting properties (Table 1).

The bioactive compounds can modulate the microbiota directly or indirectly. The mechanisms of direct interactions could include the modulation of (1) the composition (e.g., through inhibition of nucleic acid synthesis, membrane integrity, and permeability, energy metabolism [14,15,17,18,26,33]); (2) the colonization of bacteria [21]; (3) the release of bacterial metabolites (e.g., SCFA) [16,19,23,24,31,32,34]; (4) the enzyme activity of gut microbiota (e.g., *β*-glucosidase, *β*-glucuronidase activity) [19]. The indirect interactions affecting bacteria could also occur through (1) modulation of the pH in the intestine [16,31], (2) the gastrointestinal transit time [35], and (3) the synthesis and release of antimicrobial peptides [36].

It has been shown that the alcohol-soluble dietary fibre fraction of Chinese cabbage promoted the growth of intestinal lactic acid bacteria strains (*L. plantarum*, *L. casei*, *L. delbrueckii*, *L. acidophilus*) [26]. This was probably due to the effect of oligosaccharides formed during enzymatic hydrolysation because probiotic bacteria might enzymatically convert certain oligosaccharides to increase their bioavailability and further stimulate bacterial growth. Another example of waste extracts that supported the growth of the probiotic bacterium is capsicum seed cores [37]. In that study, the effect of vegetable side stream was expressed by the survival of *L. brevis* in the waste extract. That study showed that methanolic capsicum waste extract supported and enhanced *L. brevis,* a type of probiotic bacterial growth. There are also scientific data that shows the condition of the bacterial population under the stimulation of vegetable by-products. For example, diets containing carrot preparations from the root and pomace could boost intestinal microbial activity [19]. Supplementing the diet with carrot preparations containing anthocyanins in that study significantly increased the production of propionic and butyric acid in the cecum. A possible mechanism for this effect could be related to the anthocyanin-induced reduction in oxygen tension in the intestine and, thus, modulation of intestinal microbiota and SCFA production. It has been shown that anthocyanins and their metabolites may exert a positive modulation of the intestinal bacterial population, significantly enhancing the growth of *Bifidobacterium* spp. and *Lactobacillus* and *Enterococcus* spp. [38]. The microbiota-modulating effect is desired not only in the organism’s healthy conditions but also during bacterial infection; thus, wastewater as a by-product of tofu production has been tested in the aspect of *S. enteritidis* infection. Tofu wastewater includes a high concentration of organic products and *Lactobacillus* that up-regulated the relative abundance of *Lactobacillus*, counteracting alterations in the gut microbiota caused by *S. enteritidis* infection in chickens [21]. In that study, a 10% addition of tofu whey wastewater to the diet significantly increased the mRNA levels of tight junction proteins (ZO-1, claudin-1, occludin) in the intestine compared with chickens in the control group without intervention and *S. enteritidis* infection.

### 2.2. Antimicrobial Activity toward Microbial Pathogens

Taking into account the effect of vegetable side streams on microbiota, it is worth focusing deeper on its antimicrobial properties against pathogens (Table 1). The antibacterial activity of vegetable by-products can be associated mainly with carboxylic acids (benzoic, phenylacetic, and phenylpropionic), flavonoids, and other compounds, such as carotenes, glucosinolates, fibre, carbohydrates (pectin oligosaccharides and monosaccharides), sulfur compounds, and nitrogen-containing glycoalkaloids (Figure 1). The extract of onion peel, considered one of the most abundant vegetable waste, showed growth-inhibitory activity and bactericidal effect against *B.* cereus, *B. subtilis*, *E. coli*, *S. aureus*, *Salmonella* Typhimurium, *Pseudomonas aeruginosa* [14,15,17]. It has been shown that organic extracts from onion and quercetin present antimicrobial activity, and interference with quorum sensing regulated the production of violacein (antibiotic properties) and coordinated translocation of a bacterial population [39]. Furthermore, subcritical water extracts of onion peel showed antimicrobial effect against different strains of pathogenic bacteria *S. aureus* (KCCM 40510, KCCM 32395, KCCM 11335) (better than the methanol-control) by reducing cell growth by 0.7–1.1 log cfu/mL and *B. cereus* (KCCM 40935) [17,18]. Lee et al. [18] claimed that the mechanisms by which these effects occur remain to be determined and suggested that these mechanisms most likely involve the inhibition of nucleic acid synthesis, cytoplasmic membrane function, and energy metabolism.

The expanding use of vegetable fibres and their antimicrobial properties are widely described in the literature [40]. The vegetable by-products, such as tomato seeds and peels, represent an attractive, natural source of fibre [41,42] that can inhibit bacterial growth [40,43]. The extracts (ethanol and 5% acetic acid (95:5 ratio), from tomato plant residues inhibit the growth of pathogens, such as *E. coli* O157:H7, *Salmonella* Typhimurium, *S. aureus*, and *Listeria ivanovii* [44]. The authors claimed that the presence of glycoalkaloids in tomato extracts could be an indicator of antimicrobial activity. The study of Szabo et al. [42] showed that the methanolic extracts of tomato peels presented antimicrobial activity against *S. aureus* and *B. subtilis.* It has been shown that tomato and carrot by-products are good substrates for lactic acid fermentation, and interestingly, the extracts of obtained fermented by-products were characterised by antimicrobial activity toward fourteen pathogenic strains (among others *Listeria monocytogenes*, *Salmonella spp*., *E. coli*, *S. aureus*, and *B. cereus)* [45]. The authors of that study stated that lactic acid fermented vegetable by-products could be a good source of obtaining antimicrobials useful in food biopreservation. Carrot wastes are a great source of *β*-carotene (78.37 µg/g in an extract from carrot pomace; carotenoids extracted from carrot pomace using ultrasonication) [46]. Recent studies showed the antioxidant and anti-inflammatory effects of *β*-carotene and astaxanthin that may contribute to the inhibition of *Helicobacter pylori*-induced gastric inflammation [47]. Therefore, the consumption of *β*-carotene-rich foods may be beneficial to prevent Helicobacter pylori-induced gastric inflammation. Taking into account antimicrobial properties, it is worth highlighting the difficulty in fighting antibiotic-resistant bacteria, which is a challenge not only to the food industry [48]. The interesting biological properties of carrot pomace water extracts were noted that exhibited inhibitory activity against two methicillin-resistant *S. aureus* Gram+ strains and *Enterococci* (HLAR-VRE) strains [20]. 

It has been shown that other vegetable by-products from okara, cauliflower, and broccoli also exhibit antimicrobial properties [49]. In that study, okara was the most bactericidal by-product against *E. coli* O157:H7; cauliflower and broccoli showed the highest antimicrobial effect against *B. cereus*, and cauliflower was the most effective vegetable by-product against *Salmonella* Typhimurium. Analysis carried out by Gebrechristos et al. [50] revealed that potato peel extract has antibacterial as well as antioxidant properties. Based on HPLC results, they claimed that caffeic, chlorogenic, and neochlorogenic acids were the main chemical compounds found in potato peel extract, which were responsible for their antimicrobial property. According to the findings of that study, extracts from dried and milled potato peel have excellent antimicrobial efficacy for both G- and G+ bacteria. The authors of the Cheaib et al. study [20] suggested that phenolic compounds have better activity against G+ bacteria than G- ones due to their outer membrane that provides the cell with a hydrophilic surface acting as a barrier reducing the uptake, therefore imparting an intrinsic resistance of these bacteria to antimicrobial compounds [51]. The Cueva et al.’s [52] research indicates that phenolic acids (benzoic, phenylacetic, and phenylpropionic) originating from microbial degradation of different classes of dietary phenolic compounds can inhibit the growth of intestinal bacteria and pathogens at concentrations that might be physiologically relevant. The results of that study show that high concentrations of phenolic acids (1000 μg/mL) may limit the growth of *E. coli* lpxC/tolC. The antimicrobial effect of the phenolic acids against *E. coli* observed in Cueva et al. [52] was structure-dependent (i.e., benzene ring substitutions and saturated side-chain length); thus, phenolic acids seemed to show greater antimicrobial potency against *E. coli* strains than their corresponding dietary precursors, such as the flavan-3-ol monomers and dimers. The antibacterial action of phenolic acids against *E. coli* was likewise found to be concentration and bacteria-strain dependent (Figure 1). Bioactive compounds from vegetable by-products may have a different effect depending on the bioavailability. Their microbial-supporting activity, especially in the colon, could be observed because their dietary phenols, polyphenols, and tannins are poorly absorbed in the small intestine, and over 95% reach the colon as a substrate for microbial fermentation in the gut [53]. Taking into account the antimicrobial effect of vegetable by-products against bacteria depends on the type of extraction of bioactive compounds (Figure 1). Brito et al. [27] showed the difference in antibacterial activity with the same matrix but extracted with different solvents showing that not all antimicrobial phytochemicals are soluble in the same solvent, studying cabbage stalk flour essential oils.

Some of the vegetable by-products contain vitamins, such as C, B1, B2, B9, and K1 and provitamins, such as beta-carotene. It has been shown that the intake of vitamins, particularly those with antioxidant properties, can beneficially modulate the intestine microbiome. In a group of adults with cystic fibrosis was observed a negative correlation between dietary intake of vitamin C and *Bacteroidetes* genus and a positive correlation between intakes of beta-carotene equivalents, vitamins E and C, and *Firmicutes* and its lower taxa (e.g., Clostridium) [54]. In a pilot study, Pham et al. [55] claimed that colon-derived vitamins C (500 mg ascorbic acid/day), B2 (75 mg/day), and D (60 µg cholecalciferol/day) may modulate the human gut microbiome in terms of metabolic activity and bacterial composition. More vitamins affect the microbial abundance, diversity, richness, and SCFA production, which was detailed described in Pham et al.’s [56] review study.

Vegetable agro-industrial by-products, such as stems, roots, flowers, leaves, seeds, peels, and buds, are good sources of essential oils [57], which contain volatile compounds with antimicrobial properties. Among them, the essential oils of garlic and onion show an effective antibiofilm activity against *Listeria monocytogenes* [58]. According to Benkeblia [59], when seasoning is desired, essential oil extracts of onions and garlic can be used as natural antimicrobial additives in various food products. The author of that study noted that *Salmonella* enteritidis was strongly inhibited by red onion and garlic extracts. Another source of essential oils of vegetable origin is celery, containing limonene and *β*-eudesmene as the main compounds. Essential oils from celery were characterised by antimicrobial activity against *S. aureus*, *Candida albicans,* methicillin-resistant *S. aureus*, *Salmonella enterica* serovar Typhimurium, and *Listeria monocytogenes* based on a measure of IZ [60].

It is worth highlighting the role of antinutrients in vegetables that can influence the microbiota and microbiota regulating mechanisms. The antinutritional factors found in vegetable-based foods are oxalates, tannins, phytates, lectins, lignins, saponins, alkaloids, and various glycosides. The aerial plant parts may contain high amounts of oxalic acid [61], and a diet high in oxalate can increase the risk of kidney stone growth [62]. Kumar et al. [63] noted that oxalate-exposed macrophages have reduced anti-bacterial response. Oil extracts of fluted pumpkin seeds (*Curcubita pepo*) contain tannins, cyanogenic glycosides, cardiac glycosides, and saponins and showed antibacterial properties against *S. aureus* (60% IZ) and *E. coli* (50% IZ) [64]. Oil extracts of black mustard seeds *(Brassica nigra*) containing tannins and alkaloids showed antibacterial properties against *S. aureus* (40% IZ) and *E. coli* (50% IZ) [64]. Moreover, sweet ethanolic extract of potato leaves containing tannins, alkaloids, and cardiac glycosides inhibits the growth of *S. epidermidis*, *S. aureus* and *E. coli *(<12 mm-bacterial IZ). In that study, the authors noted that cassava leaves ethanolic extract was more effective in antibacterial activity against *S. aureus* and *E. coli* than ethanolic extract of cassava peels in 125 and 250 mg/mL concentrations [65].

### 2.3. Helth Benefits of Fibre from Vegetable Waste Sources

The consumption of dietary fibre has a positive effect on health since it has been related to decreased incidence of several diseases and prevention of colon cancer, maintenance of cardiovascular health, reduced serum lipid and cholesterol levels, and delayed absorption and digestion of carbohydrates [66,67,68]. Dietary fibres are the main source of carbon and energy for colonic microbes [8,69,70]. Dietary fibre consumption supports beneficiary the intestinal microbiota, which leads to risk reduction in both intestinal and systemic pathologies, which was already comprehensively reviewed by Barber et al. [4]. Most vegetable by-products are a rich source of dietary fibre which is divided into two basic categories, soluble and insoluble. Cellulose, hemicellulose, and lignin are not soluble in water, whereas pectins, gums, and mucilages become gummy in water [66]. 

In a bakery, pumpkin seeds and rinds can be employed as nutritional fibre sources [71]. Potato starch can be a source of dextrin, which reduces feed intake and causes changes in the distal intestine, which may promote intestinal health and prevent weight gain and obesity [72]. Ma et al. [73] studied the intestinal protective effect of insoluble dietary fibre from carrot pomace against heavy metal damage and highlighted the potential of the ultra-micro grinding process to produce a high added-value fibre ingredient from carrot residues. The apoptosis assessment of that study revealed that the ground insoluble fibre could effectively protect Caco-2 cells from lead ion damage and showed that carrot fibre has no toxicity for these cells (10.0 mg L(−1)). Water-soluble dietary fibre obtained from the cellulose fraction of Chinese cabbage using enzymatic hydrolysis is a potential source of dietary fibre with prebiotic, hypoglycemic, and hypolipidemic effects [26]. Often described, water-based methods of extraction do not allow to achieve the full potential of legumes by-product to be used. In the work of Gutöhrlein et al. [74], it was presented that incubating pea hulls with ethanol resulted in an increased total dietary fibre content in the extract toward the water extraction, but the hydration properties of fibre extract were similar. However, pectin acidic extraction loosened the rigid cell wall matrix, increased the amount of soluble the remaining fibre and enhanced water-binding properties. On the other hand, the chelating-assisted pectin extraction presented the highest water-binding properties due to the formation of pectic-related sodium carboxylate structures that favoured the water uptake. It was highlighted that the hydration properties of hull and husk fibre of legumes could be markedly improved using pectin-targeted chemical extraction procedures and were more effective than the ethanol-based method. 

### 2.4. Intestine and Inflammatory Bowel Diseases

Inflammatory bowel disease (IBD) is a term used to describe disorders that involve chronic inflammation of the gastrointestinal tract, microbial dysbiosis, and dysregulation of the mucosal immune system [75]. IBD includes two conditions, such as Crohn’s disease and ulcerative colitis. The management of these diseases usually requires prolonged drug interventions, which can display frequent side effects, such as headache, abdominal pain, nausea, diarrhoea, and rash [76,77]. Due to the difficult treatment of IBD and decreased quality of life scores by medications, scientists are focused on the possibility of introducing dietary components and supplementation to support the pharmacotherapy response as well as the general clinical patients’ conditions [78]. It has been confirmed that polyphenols and non-phenolic compounds exert a promising effect in the aspect of IBD.

Dietary polyphenols are widely considered by scientific research in the aspect of protective and therapeutic effects in the management of IBD. It is proven that polyphenols mediate the down-regulation of inflammatory cytokines and enzymes, enhance antioxidant defence, and suppress inflammatory pathways and their cellular signalling mechanisms [79,80]. Since by-products are defined sources of bioactive compounds, such as polyphenols, it becomes an issue of interest to obtain an extract, concentrate, or isolate of phenolic compounds to be applied in the therapeutic approach of IBD or colon cancer [81,82]. During microbial activity in the intestine, polyphenols and dietary fibre are metabolised to short-chain fatty acids (SCFA) (e.g., butyric, propionic, and acetic acid) [83]. Because SCFAs have multiple beneficial effects on the gut barrier, such as providing energy to colonocytes and the bacteria living in the gut, limiting the growth of pathogens through lowering the luminal pH of the gastrointestinal tract, increasing mineral absorption, and promoting bile acid secretion [84], they are considered one of the factors in IBD treatment. SCFA contribute to the regulation of metabolic syndromes caused by IBD and other intestinal diseases [85,86]. At the cellular level, SCFAs can have direct or indirect effects on innate and adaptive immune cell generation and function [87]. The review study by Li et al. [7] showed synergic interactions between polyphenols from plant-based food and their metabolites in the reduction in oxidative stress, inhibition of TNF-*α*, IL-6, IL-8, and IL-1*β* secretion, suppression of NF-*κ*B, upregulation of Nrf2, gut barrier protection, modulation of immune function, and alleviation of IBD symptoms. Based on their findings, Fotschki et al. [31] proposed that chicory products, particularly chicory root meal, should be regarded as a good source of dietary fibre that modulates the intestinal physiological fermentative processes and lowers colitis-related illnesses.

Onion dry skin is a common vegetable food waste and the main natural source of quercetin (flavonoid with antioxidant properties) in foods [88]. In the review study of Lyu et al. [89], it has been shown that quercetin exerts the anti-IBD effect by consolidating the intestinal mucosal barrier, enhancing the diversity of colonic microbiota, restoring local immune homeostasis, and restraining the oxidative stress response. Dietary quercetin ameliorates experimental colitis in mice by remodelling the function of colonic macrophages via a heme oxygenase-1-dependent pathway (Rag1-deficient mice, oral administration of 10 mg/kg body weight for 7 weeks). Systemic delivery of quercetin significantly reduced the severity of chronic intestinal inflammation, evidenced by histological inflammation scores, body weight/colon length ratios, mucosal wall thickness, and a loss of colonic goblet cells [90]. 

Moreover, non-phenolic compounds from plant food by-products can exert anti-inflammatory, anti-oxidative, and/or anti-dysbiotic effects and are considered an interesting tool against IBD [91]. In potential amelioration of inflammatory bowel disease treatment through artichoke pectin administration is considered [92]. There are mechanistic links between IBD pathology and vitamin/mineral deficiencies, and normalising their levels by supplementation is clinically beneficial [93]. Vegetable by-products or extracts can be a good source of some micronutrients, and some of them have the potential to be applied to IBD patients. It has been shown in Costantini and Pala’s [94] pilot study with 12 CD patients that 600–1500 mg of vitamin B1 daily completely alleviated symptoms of fatigue in 10 out of 12 patients, with the remaining two also reporting significant improvement. The ethanolic extract of onion husk contains vitamins B1 (0.53 ± 0.05 mg/100 g), B2 (0.070 ± 0.007 mg/100 g), and B9 (4.06 ± 0.41 mg/100 g) and are characterised by high antioxidant capacity [95]. Furthermore, pea by-products are rich in vitamin B1 (1.61 mg/100 g) and contain vitamins C and B2 (34.65 and 0.4 mg/100 g, respectively), potassium (25.78%), calcium (8.29%), and magnesium (6.82%) [96]. Similarly, vitamin K is insufficient in patients with IBD [97]. Shiraishi et al. [98] suggested that vitamin K exerts a protective effect against murine dextran sulfate sodium colitis and could be a potential treatment target for IBD. Vegetable wastes, such as the outer leaves of cabbage and Brussels sprouts, have been shown to provide a source of vitamin K1 and contain more than the inner leaves [99]. It has been estimated that 15% of IBD patients are affected by zinc deficiency [100]. Rempel et al. [101] showed up results within the patients with IBD (10% with Crohn’s disease and 6% with ulcerative colitis) were diagnosed with zinc deficiencies. The main sources of zinc are cereals and meat [102], but also fortified vegetables can be the source of this valuable microelement. The result of feeding mice with lettuce and komatsuna fortified with high zinc content indicates that the intake of these leafy vegetables can ameliorate zinc deficit and might be useful in protection from several diseases associated with this deficiency [103]. Preclinical and clinical research on IBD demonstrated that the administration of prebiotics, such as inulin, would enhance the gut mucosal barrier [104,105]. Bracts of artichoke could be used as a source of inulin and caffeoylquinic acids for the production of food additives and nutraceuticals [106]. Chicory plant extracts, such as root and peel ethanol extracts, are likewise high in inulin (60.1 and 46.8 g per 100 g of fresh mass, respectively) [107]. Leaf and seeds had decidedly lower mass fractions of inulin (1.7 and 3.2 g per 100 g of fresh mass, respectively) [107]. Juśkiewicz et al. [108] studied different preparations of chicory roots, such as flour, pulp, high-(IN_HM_) and low-molecular inulin (IN_LM_), and showed that chicory preparations lowered caecal pH and ammonia concentration (except IN_LM_ preparation), evoked increased hydration of caecal digesta and protein content.

*β*-galactosidase and *β*-glucuronidase are tetrameric enzymes that hydrolyse *β*-D-galactose and *β*-D-glucuronide, respectively. The excess amounts of the *β*-glucuronidase enzyme may promote a higher risk of colon cancer [109]. Quercetin diglycoside preparation from yellow onion waste from industrial peeling efficiently reduced the faecal and caecal bacterial *β*-glucuronidase activity [110]. A significant decrease has been observed in the activity of bacterial *β*-galactosidase and *β*-glucuronidase in the caecal digesta of rats fed purple carrots preparation-containing diets [19]. More information about the vegetable by-products and their effect on *β*-galactosidase and *β*-glucuronidase activity are included in Table 1.

Vegetables and their by-products are a good source of bioactive compounds, such as polyphenols and minerals, and they could be considered in the IBD, but they also present challenges, and their mechanisms of action, safety, and bioavailability need to be thoroughly understood.

## 3. Modulation of the Immune Parameters

Bioactive compounds that can be contained in vegetable side streams can modulate immune system parameters and, therefore, play an important role in the prevention and treatment of diseases. Immune response to various compounds might be systemic or local, and it is a key aspect of immunity that supervises all organs. The immunological properties can be manifested through their anti-inflammatory, antioxidant, immune-modulating, apoptosis-inhibiting, and cytoprotective (Figure 2), but also autophagy-inducing, antidiabetic, and neuroprotective effects [111]. Some of these activities will be discussed in this chapter.

### 3.1. Anti-Inflammatory, Antioxidant and Cytoprotective Effect

By-products origin from vegetable production are a rich source of phytochemicals that can induce many signalling pathways, all of which play critical roles in regulating various tissues’ immunity, especially the intestinal mucosal immunity and modulation of microbiota directly or indirectly, contributing to intestinal homeostasis and general well-being. That homeostasis can be disturbed by many factors; for example, intrinsic, such as inflammation, or external, such as environmental pollution or diet, and many others [112]. Inflammatory diseases have become a scourge of modern society. Chronic inflammation is the result of dysbiosis, increased intestinal permeability and activation of the immune system [113]. Many phytochemicals contained in side streams may exhibit cytoprotective and immunoregulatory properties. Some vegetable by-products, such as pomace, skin, husk leaves, and stems, contain several bioactive compounds and, in addition, they differ in chemical properties, such as solubility, polarity, and stability, so their simultaneous recovery is a challenge. To recover and maintain their immunomodulatory properties, various technological solutions are tested. Kabir et al. [2] reported that hot-water extraction of bioactive components (polyphenols) was considered to be effective (in comparison to the ethanol-based one) when performed for different vegetable production wastes, including broccoli leaves, broccoli stems, asparagus stems, cabbage outer leaves, Chinese cabbage outer leaves, lettuce outer leaves, cowpea hull, and black azuki bean hull. In that study, waste extracts were compared in terms of their antioxidant, immunoreactive, and cytoprotective activity. In the reported Kabir et al. [2] study, the total phenolic concentration was measured using the Folin–Ciocalteu assay and free radical scavenging activity with the DPPH assay was tested. The cell viability impact of extracts was tested on the in vitro model of African monkey kidney cells (MA 104) stimulated with 2,2′-Azobis(2-amidinopropane) dihydrochloride (AAPH), which is a chemical compound used to study the chemistry of the oxidation of drugs. Most extracts of vegetable wastes and by-products exhibited potent antioxidant activities in DPPH free radicals and AAPH peroxyl radicals. The strongest antioxidant potential was shown by broccoli and cabbage outer leaves extracts (EC50 = 430.8–476.1 l and 437.0–914.8 μg/mL, respectively), which was proportional to the total phenolic concentration (TPC = 23.5–25.5 and 21.1–22.1 mg ChAE, respectively). Water-based extracts of broccoli waste showed high cytoprotective activity (>80%), reflected by cell viability against AAPH-induced cytotoxicity (89.9 ± 1.0 6%). However, the highest cytoprotective and, in principle, the proliferation-stimulating effect was expressed by the black azuki bean hull extract (210.9 ± 11.5% of cell viability). That was surprising because Xu and Chang observed an antiproliferative effect of the black azuki bean extract and its hull but also broken and damaged seeds water-extract [114]. That effect was reported in a dose-dependent manner against all digestive system cancer cell lines (CAL27, AGS, HepG2, SW480, and Caco-2), ovarian cancer cell SK-OV-3, and breast cancer cell MCF-7. Although the observed effect was the opposite (cytotoxic) of the one reported by Kabir et al. [2], the same phytochemicals (procyanidins, saponin, and phytic acid) were reported to be responsible for the regulation of proliferation, cytoprotection, and cytotoxicity. The reported difference in the biological effect of the same phytochemicals extracted from the identical side stream may be mainly due to the applied biological model and the origin of the cell lines (healthy vs. cancer-derived) used for testing. It may also be caused by the occurrence of other phytochemicals, such as phytates, in that extract. Raw material derived from legumes is a rich source of phytates, and water extraction can be equally effective in their recovery. What is important, phytate’s biological and cytoprotective activity differs depending on the type of tissue, which was reported by Markiewicz et al. [115]. She tested the role of the combined effect of 1 mM phytate and 1 mM butyrate on cell lines derived from cancer (HCT116 and HT-29) and healthy (NCM460D) human colonic epithelium. It was observed that the combination of phytate and butyrate disturbed the proliferation and cell cycle and triggered apoptosis and/or death in both studied cancer colonocytes to a greater extent compared to healthy colonocytes. The expression of TNF-*α* and IL-8 was boosted in cancer-origin lines. However, in healthy colonocytes, phytate activated the survival pathway without stimulation of inflammatory response (reduced expression of immune response factor, NF-*k*B1, IL-8, TNF-*α*, and reduced cell proliferation and oxidation markers, PTEN, iNOS). This could imply that healthy colonocytes’ response to phytate protects colonic epithelium from the loss of integrity and tightness that would occur if inflammation developed. The cell line origin is a particularly important criterion in assessing the mechanism of action of phytochemicals, especially those obtained from by-products. It is also important to adjust the method of extraction and processing of by-products when we think about the usefulness, bioactivity, and bioavailability of particular biologically active compounds from side streams. In the mentioned study by Kabir et al. [2], it was reported that the cytoprotective and free radical scavenging potential was higher in water broccoli extracts than in ethanol-contained ones. However, when we compare the effectiveness of the recovery of another cytoprotective component from broccoli by-products, which are amino acids, water-based extraction can be ineffective in terms of extracting polar amino acids. It was reported by Drabińska [116] that methods of extraction based on alcohol-containing solvents (methanol-based solvents used for analytical purposes) are more effective, especially for free amino acid recovery. It was reported that doubling the percentage of methanol results in the 14-times higher efficiency of the amino acids extraction. However, some extraction methods will not always be suitable for implementation in the food industry, which is why Drabińska [116], in her study on broccoli by-product adaptation, indicated that effective fortification, especially in polar and exogenous free amino acids, can be favourably influenced by the incorporation of powdered lyophilised broccoli leaf instead of their water extracts, in which, e.g., proline and glutamine were lost. The increased concentration of alanine, serine, aspartic, glutamic acid, leucine, threonine, valine, and lysine was also noted what was the added-value to gluten-free products (sponge cake and pasta) that were fortified with broccoli by-product [116,117].

Amino acid fortification of gluten-free products seems to be crucial since they have a multifaceted influence on the physiology of the body through the formation of protein structure, detoxification processes, and regulation of anabolic and catabolic metabolism. Several amino acids, including L-arginine, L-methionine, L-glutamine, and taurine, were found to protect the intestinal epithelial cells (in vitro-Caco-2 cell line and in vivo and ex vivo, Wistar rats models) from the local toxicity caused by a drug absorption enhancer, sodium laurate (C12) [102]. Endo et al. [118] proved that the cytoprotective action of amino acids against the local toxicity was caused by the induction of biosynthesis of heat shock protein 70, a decrease in intracellular concentration of Ca^2+^, and suppression of histamine release. The action of amino acid fortification can, therefore, be local as well as systemic. It was also demonstrated that variations in amino acid content and abundance are compartment-specific in the gastrointestinal tract. [119]. Its metabolism can influence the profile of amino acid-metabolizing bacterial communities and induce mechanisms of macrophages and dendritic cell activation via toll-like receptors (TLRs), NOD-like receptors, and autoinducer-2. Dietary amino acids, directly and indirectly, regulate the gut-microbiome-immune axis via aryl hydrocarbon receptor, serotonin/5-hydroxytryptamine, and other signalling pathways, all of which were comprehensively reviewed [120].

The significant role of proteins and amino acids in dried tomato processing waste was also described by Nour et al. [121]. It was reported that samples contained 176.2 g/kg protein, 21.9 g/kg fat, 524.4 g/kg crude fibre, and 42.1 g/kg ash. The high content of the essential amino acids represented 34.2% of total protein, with the most abundant being leucine, lysine, and isoleucine. That by-product was also rich in unsaturated fatty acids, representing 77.04% of the total fatty acids, linoleic being the major one. Ellagic and chlorogenic acids were the most abundant phenolic acids. What is more, it was confirmed that dried tomato wastes contained considerable amounts of lycopene (510.6 mg/kg) and *β*-carotene (95.6 mg/kg) and, thus, exhibited good antioxidant properties. Lycopene, one of the most potent natural antioxidants, has been extensively studied for its cancer-preventive and anti-inflammatory properties [122]. In the mentioned study by Cha et al. [122], lycopene (10–30 µM) in a dose-dependent manner reduced the expression of inflammatory markers in the lipopolysaccharide (LPS)-stimulated human colorectal cancer (SW 480) cell line. In cells treated with lycopene and LPS, the mRNA expression of proinflammatory markers, such as TNF-α, IL-1*β*, and IL-6, induced nitric oxide synthase (iNOS) and cyclooxygenase 2 (COX-2) decreased significantly. The concentrations of prostaglandin E2 (PGE2) and NO decreased, but the expressions of NF-*κ*B and c-Jun N-terminal kinase (JNK) on the protein level also decreased significantly according to lycopene concentration. Lycopene works cytoprotective but also immunoregulatory on whole blood cells, which was reported by Hazewindus et al. [123]. In that study, lycopene was found to enhance the release of the immunoregulatory cytokine IL-10 in whole blood cells. Moreover, the combination of lycopene, ascorbic acid, and *α*-tocopherol tended to display a synergistic effect on IL-10 secretion. Lycopene also effectively reduced inflammation and inhibited the production of TNF-*α*. Neither ascorbic acid nor *α*-tocopherol potentiated nor reduced the anti-inflammatory effect of lycopene on TNF-*α* production in that model. The promising effect of the combination of that compounds manifest through the boosting of IL-10 secretion. The cytoprotective effect of tomato pomace as a side stream was also tested by Abbasi-Parizad et al. [124]. It was reported that tomato pomace ethanol extract demonstrated 62.8% inflammation inhibitory activity at 25 µg/mL dose to the positive control (Caco-2 cells induced with IL-1*β*). A significant reduction in IL-8 expression after tomato pomace extract treatment and increased survivability of Caco-2 cells have been noted. Abbasi-Parizad et al. [124] reported that phenolic acids extracted from tomatoes mainly consisted of cinnamic acid, p-coumaric, and caffeic acids. Flavonoids represented 65% of TPC in tomato pomace, and naringenin and naringenin chalcone were the most represented ones. Total polyphenol content in tomato waste can vary depending on such factors as variety, size, cultivars, degree of ripening, and industrial process procedures [42]. 

### 3.2. Anti-Allergic and Innate Immunity Inducing Effect

The medical problem of food allergy and its complications is constantly growing and affects about 11–26 million people yearly [125]. Thus, the global costs of allergy therapy and diagnostics are increasing (2.69 billion USD in 2018, with an estimated increase of 7.4% to 2028). It seems that some bioactive compounds from side streams can be promising therapeutic and preventive factors. Flavonoids, including luteolin and apigenin, but also fisetin and quercetin, present in high doses in waste from the production of plants of the Amaryllidaceae family, are reported to be immunoactive [126]. Their broad biological activity induces beneficial immunological reactions independent of the aforementioned processes. It has been proven that reported flavonoids, including luteolin, apigenin, and fisetin, are inhibitors of IL-4 synthesis and CD40 ligand expression by humans basophils [127,128] and revealed anti-allergic effect in a BALB/c murine model of allergic asthma and rhinitis [129]. As Hirano et al. [127,128] reported, flavonoids can suppress the activation of Activator protein 1 (AP-1), which, with the Nuclear factor of activated T cells (NFAT), are important transcription factors in the regulation of secretion of various cytokines, including IL-4 as well as of CD40 ligand expression. Luteolin, apigenin, and fisetin inhibit IL-4 and IL-13 production with an IC50 value of 2–6 µM, histamine release, andCD40 ligand expression but do not suppress leukotriene C4 production by activated basophils in response to cross-linkage of the high affinity of IgE receptor (FceRI) and IL-3. Luteolin was also reported to not suppress phosphorylation of spleen tyrosine kinase (Syk) or tyrosine–protein kinase (Lyn), nor did it suppress different pathways of MAPK activation in KU812 basophilic cell line, stimulated with Calcium Ionophore A23187 and phorbol 12-myristate 13-acetate (PMA). However, it inhibited the phosphorylation of c-Jun and DNA binding activity of AP-1 in nuclear lysates from stimulated (A23187 + PMA) cells. On this premise, they developed a hierarchy of the allergy-inhibitory activity of flavonoids. It was also stated that despite the chemical and structurally similarity to fisetin, luteolin, and quercetin, another flavonoid, myricetin, failed to inhibit CD40 ligand expression in KU812 cells in the same concentration (10–30 μM). These observations were verified by Jang et al. [129] group in in vivo studies in a mouse model. They reported that OVA-intraperitoneal immunised mice treated with 100 μg/kg of luteolin before the intranasal challenge manifested a decrease in IL-4, IL-5, and IL-13 in their lung homogenate compared to control groups and also showed a significant decrease in inflammatory cell infiltration (including eosinophils, neutrophils and lymphocytes in bronchoalveolar lavage fluid) after luteolin treatment [129]. Those findings may contribute to the clinical application for allergic patients of selected flavonoids as an alternative and complementary therapeutic agent to the preventative strategy for allergic diseases through diet management, and taking into account the possible sources of their obtaining, it can be involved in a sustainable waste management strategy. 

Another bioactive ingredient present in many different side streams is polysaccharides fraction. Many studies on the immunogenic properties of these compounds are concerned with white asparagus skin, old asparagus stems, and asparagus sections, whose production reaches almost 4200 tons per year [130,131]. These by-products may be a valuable source of functional ingredients for the food sector thanks to Wang [130,131], who reported immunogenic properties. It was reported that asparagus skin is rich in a pectic-like structure with a relatively low degree of esterification with structurally dominated sugar residue, 4-*α*-D-GalpA (39.7 mol%), while other residues, including *α*-L-Araf,3-*α*-L-Rhap, 2,4-*α*-L-Rhap, and 4-*β*-D-Galp, were also detected with a comparable amount. That polysaccharides composition (tested in doses 50–200 μg/mL) modulated the immune response of RAW 264.7 macrophages by increasing the release the cytokines (proinflammatory IL-6, TNF-*α,* and anti-inflammatory IL-10) and improving the expression of mRNA and phagocytic activity, one of the most significant characteristics of macrophage activation is phagocytic activity, which is an important barrier for the host to defend against the innate immune system. The antigen-presenting cells that derived from the macrophages after phagocytosis can interact with lymphocytes to regulate adaptive response. That is why the report by Wang et al. [130], boosted by about 20% phagocytic activity of macrophages, was promising. That effect was reported for 50 μg/mL of white asparagus skin water extract dose, whereas Nie et al. [132] reported that the 200 μg/mL polysaccharides extract from lettuce stems that contain sulphate radicals can be even more promising because it promotes macrophage proliferation without cytotoxicity. It stimulated phagocytosis and nitric oxide production. The results suggest that polysaccharides from side streams of the food industry could be explored as immunomodulatory agents in the field of pharmaceuticals and functional foods.

Another material, which due to many contained phytochemicals and their health benefits, can be a technological challenge is carrot pomace. As in the case of lyophilised broccoli leaf, in order not to lose any of the bioactive ingredients, the carrot pomace can be subjected to freeze-drying, followed by encapsulation [133]. In that form, it was introduced to the yoghurt formula. Yoghurt with carrot-pomace fortification, in addition to immunostimulating proteins, peptides, and volatile fatty acids, also provides beta-carotene and is recommended for daily intake. It was proven that beta-carotene is poorly soluble in water but also chemically unstable and prone to oxidation in the presence of acids, light, heat, oxygen, and metal ions, so encapsulation seems the best solution to incorporate it into the product in active form. Cañete et al. [134] discussed the potential immunomodulatory effect of this phytochemical from the native form to its metabolites and indicated that retinoic acid is the most potent in this regard. After consumption, the enzyme splits beta-carotene into two trans-retinal molecules, which are either oxidised into retinoic acid by retinal dehydrogenase or reduced into retinol by retinal reductase. Cañete et al. [134] reported that retinoic acid is involved in the homeostasis regulation, T cell differentiation, movement of T cell migration into tissues, antibody-dependent T cell development, proliferation and differentiation of B cells, B cells protection from apoptosis by binding with toll-like receptor 9 (TLR9), modulation of granulocyte, neutrophils differentiation, and treatment of cancer. Moreover, it serves as a signalling molecule in hematopoiesis during the embryonic stage and in vascular development. That is why Šeregelj et al. [133] concluded that encapsulated carrot pomace could be used for the development of bioactive, fortified yoghurt and, potentially, other products. 

The encapsulation was also a promising way of the distribution and preservation of bioactive compounds of beetroot side streams [135]. Betalains, which consist of yellow betaxanthins and red betacyanins, are found mainly in peels (54%), crowns (32%), and, to a lower extent, in the flesh (14%) [136]. Other relevant by-products from beetroot are the stems and leaves, which have valuable nutritional qualities and bioactive compounds. The beetroot post-harvest treatment products were analysed in depth, which proved that they contained high amounts of phenolic compounds and betalains but also carotenoids, minerals, vitamins, carbohydrates, and amino acids. It was referred that this phytochemical-rich waste possesses antimicrobial and cytotoxic activity and its usage contributed to relevant functional food [136,137,138]. For the cytotoxicity of side stream compounds in Vulić et al. [137], studies of human tumour cell lines MCF7 (breast adenocarcinoma) and MRC-5 (fetal lung) were used. In the sulphorhodamine B (SRB) assay, it was shown that the cytoprotective/cytotoxic properties of beetroot pomace 50% aqueous ethanol extract were manifested in a dose-dependent manner. The concentration <125 µg/mL worked cytoprotective, but doses of 125–1000 µg/mL showed a cytotoxic effect. It was also presented that beetroot pomace waste, after industrial food processing, exhibited good antioxidant properties by effectively scavenging DPPH, hydroxyl, and superoxide anion radicals. Beetroot material was found to be bioactive also in terms of its above-mentioned antibacterial properties. Thus, its immunoactive effect can be considered on many levels and can be the result of direct or indirect (through microbiota modulation) action. In addition to all that effect, Vulić et al. [138] also proved that the hepatoprotective effect of beetroot extract in Albino Wistar rats had been induced with Carbon-tetrachloride (CCl4), which was concluded to be caused by suppressing the harmful effect of excessive production of free radicals. 

### 3.3. Impact on Various Aspects of Livestock Immunity

For many years, the vast majority of food industry side streams products have been used as feed. In addition to improving production efficiency, it is expected that such additives can improve meat quality and animal immunity, among other things. One of the side streams that has been successfully applied to improve livestock immunity was the asparagus by-product [139]. In these studies, the cytoprotective effect on the intestinal epithelial cells and a beneficial effect on the intestinal microbiota profile of broiler chicks were reported after the diet supplementation with trimmed asparagus by-products. Nopparatmaitree et al. [139] reported that 10–50 g/kg supplementation of broiler chicks (Ross 308^®^) with trimmed asparagus by-products significantly increased LAB and *Enterococcus* spp. quantity, as well as acetic, propionic, butyric, and total SCFA levels. It also significantly decreased *Salmonella* spp. and *E. coli* contents in the cecum compared with the control group (on a standard diet). Moreover, supplementation increased villus height in the duodenum and jejunum, cryptal depth in the jejunum and ileum, and villus surface areas in the duodenum, jejunum, and ileum, which contributed to increased feed intake and average daily gain of broilers. Furthermore, supplementation decreased the atherogenic index and thrombogenicity index of meat through the decrease in cholesterol content, palmitic acid, oleic acid, saturated fatty acids, and monounsaturated fatty acid levels, but it did not affect overall characteristics, pH, colour, and water-holding capacity of the chicken meat. The application of asparagus racemosus ethanolic root extracts was reported also to be beneficial in *Labeo rohita* fish feeding [140]. Dietary inclusion of asparagus extracts (50–150 mg/kg) significantly enhanced the biochemical responses of fish serum parameters manifested by glucose, total protein and albumin concentration, and alkaline phosphatase activity. It also influenced innate immune responses through the superoxide anion, lysozyme, and myeloperoxidase’s increased activity and increased mRNA expression of immune-related genes (IL-1*β* and IFN-γ) on the 15th day of treatment, whereas that effect was mitigated on the 30th day. The survivability of animals against *Aeromonas hydrophila* infection was significantly higher in fish fed with the 100 mg/kg dose of ethanolic extract. That dose seemed the most optimal for animals and other life parameters. Another side stream that is frequently used as animal feed is potato peel extract, which is rich in polyphenolic compounds but also anti-nutritious glycoalkaloids. As a feed, it was reported to have a beneficial effect on the sperm immunity of rabbit bucks under an intensive breeding system [141]. Two doses of potato peel extract (25 and 50 mg /kg) were tested on V-Line rabbit bucks. The results showed that treatment with both doses of the extract significantly improved sperm cell concentration and viability and decreased the percentage of dead spermatozoa compared to the control. Serum immunoglobulin M concentration was also significantly higher after treatment, and seminal plasma thiobarbituric acid reactive substances significantly decreased inversely proportional to the total antioxidant capacity levels, catalase, glutathione peroxidase, and superoxide dismutase activity. However, the low dose of the extract was more effective than the high one. 

### 3.4. Immunoactive Properties of Liquid Post-Production Wastes

The production of food based on vegetable raw materials is also associated with the formation of liquid waste. It is emphasised that potato wastewater treatment especially requires extensive optimisation because of its high concentrations of organic compounds and concentrated emissions of over 20 million cubic meters of wastewater annually [7]. Potato wastewater is a rich source of starch, whose biological activity was referred to in a separate section. Beyond the role of starch, Kowalczewski et al. [142] showed that potato juice has a cytotoxic effect triggered by the synergistic effect of glycoalkaloids contained in the juice, which was not mitigated by the cytoprotective activity of phenolic acids. In that study, the fresh potato juice derived from three edible potato varieties, an industrial side stream resulting from starch production, partially deproteinised juice derived from a feed protein production line, and three different potato protein preparations were tested. All the liquid preparations were subjected to simulated digestion in the artificial gastrointestinal tract, and their influence was determined in cancer and normal human cells derived from the digestive system. The normal small intestine HIEC-6 cell line, colon mucosa CCD 841 CoN, gastric carcinoma AGS, Hs 746T, and colorectal adenocarcinoma Caco-2 and HT-29 were submitted to treatment with glycoalkaloids (1–20 μM), phenolic acids (10–200 μM), and digested potato juice and products of its processing (0.1–20 mg/mL) for 48 h. The results showed that all of the examined cell lines responded with increased cytotoxicity to glycoalkaloids in a concentration-dependent manner. In all tested cell lines, chaconine was characterised by higher cytotoxicity than solanine, which was expressed by approximately twice the lower IC50 values. Contrarily, phenolic acids (caffeic, ferulic, and chlorogenic) in the treated cell lines did not exhibit cytotoxicity. The nonlinear relation between cytotoxic potency and glycoalkaloid content was reported in all tested potato juice products. Deproteinised juice, a byproduct of industrial potato juice processing, stands out among the investigated fresh juice products for having the highest effective activity. Additionally, this preparation showed a higher cytotoxicity ratio for cancer cells than for healthy tissue-derived cells. Fresh juice and the products of its processing, subjected to gastric digestion, revealed cytotoxicity against stomach cancer AGS and Hs746T cells. Gastrointestinally-digested products revealed cytotoxicity to intestinal cancer HT-29 and Caco-2 cells but, unfortunately, also to normal CCD 481 CoN and HIEC-6 cells. On the one hand, such results are promising in the context of cancer therapy; on the other hand, they are evidence of the anti-nutritional effect of potato glycoalkaloids, which problem has not been solved yet. Other proven biological effects of liquid side stream after potato production was reported by Bzducha-Wróbel et al. [143]. That group reported that the application of deproteinated potato juice water and glycerol could increase the biosynthesis of *β*(1,3)/(1,6)-glucans of the cell wall of the yeast *Candida utilis* ATCC 9950 strains. The observed overproduction of this beneficial polysaccharide is significant since it serves as a biological modifier of the immune response, stimulates antimicrobial and anti-inflammatory activities, adsorbs mycotoxins, decreases the fraction of LDL cholesterol, exhibits anticancer, antimutagenic, and antioxidant properties, and promotes wound healing, as was reported in comprehensive work on this subject written by Chen and Seviour [144]. Despite the fact that liquid post-production waste from potato raw material has modulatory potential due to the anti-nutritional effect of glycoalkaloids, it remains a technological challenge.

## 4. Final Remarks

It is important to point out that society is facing civilisation diseases, and all ways to support health should be further explored, especially in the aspect of supporting intestine microbiota and the immune system in vitro and in vivo studies. Side streams of vegetable processing and their bioactive compounds application should be inextricably linked with studies on bioactivity, nutritiousness, and food safety. Most health concerns are associated with naturally occurring antinutritional factors, fungal and/or bacterial toxins, and environmental pollutants. According to WHO estimation, almost 1 in 10 people in the world fall ill after eating contaminated foods, and 420 thousand die every year; that is why access to sufficient amounts of safe and nutritious food is key to sustaining life and promoting good health [145]. 

Vegetable by-products are a source of many health-promoting compounds, as it was described above, but the side stream itself may also contain anti-nutritional substances. There have been some concerns raised about whether vegetable matrices are beneficial for human health because of the anti-nutritional factors, such as phytic and oxalic acid, saponin, or glycosides. For example, legume lectins are responsible for increased intestinal permeability, bacterial overgrowth and bacterial translocation [146]. Plant-derived goitrogens are another set of compounds which antinutritional properties. The term ‘goitrogen’ broadly refers to agents that interfere with thyroid function, and vegetables of the *Brassica* family (i.e., kale, Brussels sprouts, cabbage, turnip greens, Chinese cabbage, broccoli) contain goitrogens, which may decrease thyroid hormone production [147]. Legumes, such as black beans, pinto beans, kidney beans, soybeans, peanuts, and lentils, can contain phytates (phytic acid). Previous studies have reported that a phytate/iron ratio greater than 1:1 has a negative effect on iron bioavailability [148]. López-Moreno’s review [149] compiles scientific evidence regarding the physiological impact of some antinutrients (lectins, goitrogens, phytates, and oxalates) on human health and their negative effects and the culinary and industrial procedures to reduce their presence in foods. To minimise the anti-nutrient content, several processing procedures and technologies, such as fermentation, germination, de-branning, autoclaving, and soaking, are used [150].

Fungal secondary metabolites are one of the most difficult problems to solve because mycotoxins are pretty resistant to degradation and detoxification technologies. The most common vegetable fungal toxins can be divided into four major categories: ochratoxin A; patulin; trichothecenes; and Alternaria toxins that are produced by mycotoxigenic fungi (*Aspergillus*, *Penicillium*, *Fusarium* and *Alternaria*, respectively) [151]. Ochratoxin A is common in red peppers, tomatoes, and eggplants [152]. Patulin was reported for red and green pepper, tomato, onion, and cucumbers but also for juices containing carrots and tomatoes [153]. Trichothecenes were reported in potato side streams [154]. *Alternaria* toxins are common in tomato material [155]. The adverse effects that can be caused by the consumption of mycotoxins are either acute or chronic, for example, carcinogenic, teratogenic, and immunosuppressive [156]. As it was presented, side streams are not only opportunities but also associated with risks. 

Following the appropriate procedures and applying technological methods to avoid by-product spoilage, some of them can be included. The process of fermentation of plant-based products can be a natural, widely approved and, at the same time, effective method of preserving and detoxifying plant materials. Fermented foods have been shown to play an important role in microbiome diversity and concomitant immune tone in the host [147,148,149,150,151,152,153,154,155,156,157,158,159]. In the context of obtaining bioactive compounds from side streams, this is an extremely desirable feature. Due to high enzymatic activity, microorganisms can decompose and detoxify many compounds present in various types of raw materials (e.g., tannins, phytates, polyphenols, and trypsin inhibitors) [160,161,162]. Chibuike et al. [160] investigated the decrease in the contents of anti-nutritional compounds with an increasing fermentation period of maise flour. That study showed that the reductions in the anti-nutritional factors are more effective in the fermentations set-ups by LAB than under spontaneous fermentation. Adeyemo and Onilude [163] showed that the use of alpha-galactosidase enzyme by *L. plantarum* from local food sources is thus shown to reduce anti-nutritional factors in soybeans. Some vegetable by-products can contain fungal metabolites, but on the other side, others can mitigate their negative effect, such as oil from pumpkin seed, showing the antioxidant effect against the oxidative stress-inducing potential of aflatoxin [164]. 

The nutrient content in by-products depends on the part of the vegetable from which it is obtained. Roldán et al. [165] showed that by-products with a higher content of outer parts of onion (paste and bagasse) showed higher antioxidant activity than juices. The results of Liu et al. [166] study demonstrate that broccoli leaves had higher antioxidant activity, total phenolic content, concentrations of carotenoids, chlorophylls a and b, and vitamins E and K1 compared to florets or stem. Masood et al. [167] proved that the outer dry layers and fleshy peels of onion exhibited higher phenolic content and antioxidant activities compared to the inner bulb. The results of that study promote the use of vegetable parts other than the edible mesocarp for several future food applications rather than these being wasted.

Preservation treatments of vegetable by-products, such as sterilisation, pasteurisation or freezing, would have to be carefully chosen, as they are one aspect that determines the safety and stability of bioactive compounds. These methods have to be tailored to specific compounds or desirable properties. For example, sterilization may result in caramelised compounds in onion by-products that have been heat-treated [165]. It may have an impact on their nutritional composition by causing a significant loss in bioactive composition. Roldán et al. [165] claimed that pasteurisation as a mild thermal treatment would represent the better choice to stabilise onion by-products, mainly maintaining their intact bioactive composition. In the same mentioned study, the authors pointed out that stabilising onion by-products through sterilisation would provide better antibrowning properties than pasteurisation or freezing. 

From a technological point of view, the use of some vegetable by-products, such as skin, husk and pods, are limited as health-promoting preparations due to their low solubility and, thus, bioavailability [168]. Some vegetable side streams can indirectly affect our organisms through the technological and nutritional improvement of the quality of food. Liao et al.’s [169] study showed that dietary supplementation of garlic straw powder could improve the meat quality and antioxidant capacity of yellow-feathered broilers without affecting growth performance and intestinal mucosal morphology. There is a positive correlation between antioxidant activity and the polyphenolic content of vegetable extracts, affecting the degree of scavenging effects [170]. Salami et al. [171] confirmed that vegetable peel extracts, e.g., from pumpkin, can be used as a natural antioxidant in edible oils [171]. Onion husk was also characterised by high antioxidant capacity [95]. The results of Ahmed et al. [172] suggest that red onion peels can serve as a convenient and cost-effective source of high-value antioxidant nutraceuticals for protection against oxidative stress-related disorders. The use of oils obtained from the vegetable industry in bakery products can improve their nutritional quality because they have a high antioxidant activity rich in phenolic compounds [173]. Another technological challenge is the extraction of bioactive compounds, and the effects were strongly dependent on the solvent used for the extraction as well as on the extracted residue. The technological parameters, e.g., light, pH, oxygen content, and environmental temperature of some bioactive compounds, such as polyphenols, can all have an impact on their stability and, thus, poorer bioaccessibility after maintenance inside epithelial cells once they reach epithelial cells during passage through the intestinal barrier.

The authors of that review pointed out only a part of the aspect connected with health-promoting properties such as supporting microbiota, intestine milieu and immune system. However, there are many others that prevent high-fat diet-induced obesity by altering the markers important to lipid metabolism, such as the seed oil of squash (*Cucurbita maxima*) [174]. Extracts of flavonoid and saponin from black bean seed coatings at the same concentration (5 mg/mL) to stigmasterol (plant sterols, having a major function to maintain the structure and physiology of cell membranes) strongly inhibited cholesterol micellisation (correlated to the inhibitory effect of cholesterol micelle solubility as an approach to a potential reduction in cholesterol absorption) up to 55.4 ± 1.9% [175]. 

## 5. Study Design

We performed a narrative review of the current literature. The production process’s side streams are understood to be any outcomes of a production process that is not the main product of the process. Production side streams consist of by-products, non-hazardous, and hazardous waste. A by-product is considered not to be waste and includes pre- and post-harvests.

The literature search on the Web of Knowledge research platform was performed, searching all databases using the individual and collective searching of the following terms: ‘vegetable’; ‘by-product’; ‘waste’; ‘side stream’; ‘discards’; ‘health benefits’; ‘antiinflammatory’; ‘anti cancerogenic’; ‘immunomodulatory’; ‘immune system’; ‘cytoprotective’; ‘antioxidant’; ‘antibacterial’; ‘microbiota modulatory’; ‘bacteria’; ‘intestine’; ‘gut barrier protective’; ‘hulls’; ‘leaves’; ‘stalk’; ‘pomace’; ‘pulp’; ‘peels’; ‘husk’; ‘leaves; and ‘residues’ in the topic, title keywords or main text to identify articles published up until 2023. We have narrowed the search mainly to families of vegetables, such as *Amaryllidaceae*, *Amaranthaceae*, *Apiaceae*, *Asteraceae*, *Brassicaceae*, *Cucurbita*, *Fabaceae* and *Solanaceae*. Only the articles written in English, with restriction to the date of publication from 2000, have been considered. The Figures were created with BioRender.com. 

## 6. Conclusions

Undoubtedly, the amount of vegetable-origin waste is increasing along with the food demand of the growing population. Vegetable by-products are commonly considered waste, but many studies have reported that their native form, oils, or extracts can contain compounds that exert a beneficial health-orienting effect. The biological activities of side streams from vegetable processing could be successfully used, especially in the prevention of lifestyle diseases, which are based on disorders of the intestinal milieu, including dysbiosis and immune-mediated diseases resulting in inflammation. To ensure the safety of vegetable by-products, it is necessary to study the stability of phytochemicals and prevent microbiological decay to avoid the production of toxic bacterial and fungal metabolites, as well as investigate pesticide residues. Similarly, the absence of allergens, heavy metals, and organic contaminants should also be tested. The presence of antibiotics in the recycled biowaste should be carefully checked because it could contribute to the spread of antibiotic-resistant micro-organisms. The stabilisation/preserving treatments applied would have to be carefully selected to maintain the safety and stability of vegetable bio-active compounds from by-products during their whole shelf-life. Some processing technologies, such as fermentation, germination, de-branning, autoclaving, and soaking, would be applied to reduce anti-nutrient content. Future studies are needed, wherein green technologies, such as water, ethanol, and/or steam extraction, supercritical CO_2_, and others, can be adopted for the extraction from vegetable side streams. Bioactive compounds derived from vegetable wastes have the potential to be included in functional foods; however, a wide gap still exists to exploit the potential applications in nutraceuticals, food additives, and used as pharmaceutical excipients. These ingredients with natural origin may be a partial substitution for synthetic and expensive additives.

## Figures and Tables

**Figure 1 molecules-28-04340-f001:**
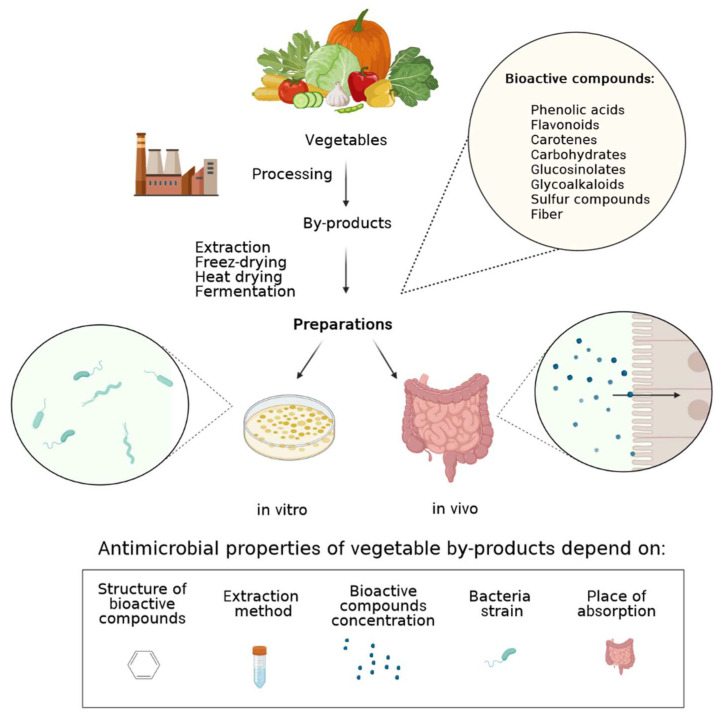
The main bioactive compounds in vegetable by-products and factors influencing the antimicrobial properties.

**Figure 2 molecules-28-04340-f002:**
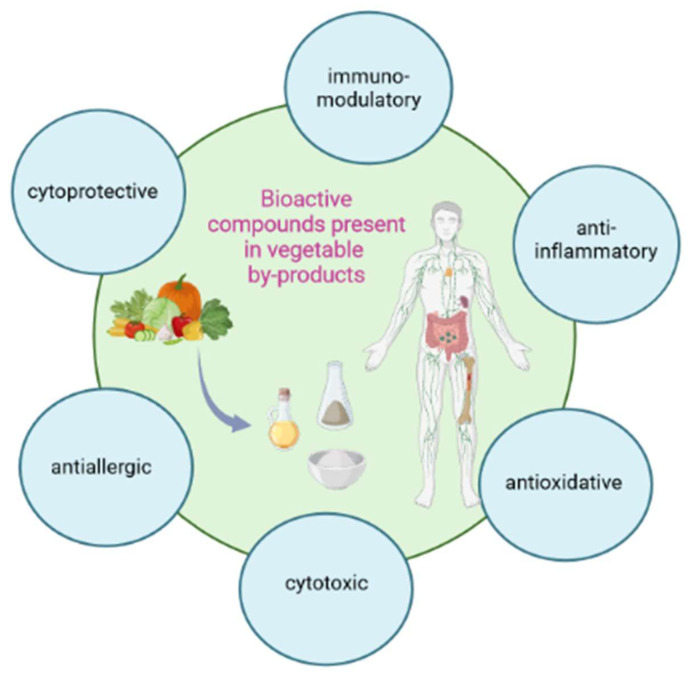
The effect of bioactive compounds present in/derived from vegetable by-products from the food industry.

**Table 1 molecules-28-04340-t001:** The effect of side streams of vegetable processing on the microbiota.

Side Streams of Vegetable Processing	Bioactive Compounds	Methods	Activity/Observations	Mechanism	Reference
Onion (Allium cepa):					
Red onion peels	Polyphenols, including flavonoids	A: Extracts (ethyl acetate, n-butanol ethanol, methanol, and water) using the maceration method. B: Antimicrobial activity-disc diffusion method; MIC; MBC and MFC.	Antifungal and antimicrobial properties. Properties were tested against G+ (*S. aureus*) and G− (*E. coli* and *Salmonella* Typhimurium) bacteria and fungi (*Aspergillus niger*, *Candida albicans*). The extracts were more active against *S. aureus* as compared to *E. coli* and *Salmonella* Typhimurium. All tested microorganisms were sensitive to studied extracts.	D	[14]
Yellow onion peels		A: Extract (ethanol–water). B: Antibacterial susceptibility—Kirby Bauer disk diffusion method and MIC determination.	Antibacterial activity against *B. subtilis*, *E. coli*, *S. aureus*, *Salmonella*, *Pseudomonas aeruginosa*.	D	[15]
Yellow onion (‘Recas’ cultivar) paste	Fructans, starch, FOS and flavonoids including quercetin	A: Two derived fractions: extract (water/ethanol soluble) rich in FOS (7%) and onion dry residue (3%). B: Rats fed (4 weeks) with an onion by-product powder (10%) and two derived fractions, extract rich in FOS (7%) and onion dry residue (3%) in the diet. Control rats fed with a control diet. SCFA—capillary electrophoresis with indirect UV detection.	The onion by-products as well as the soluble and insoluble fractions had prebiotic effects as evidenced by decreased pH, increased butyrate production, and altered gut microbiota (BGL and GUS) enzyme activities in the caecal contents.	D, ID	[16]
Orange onion peels		A: Extract (using subcritical water extraction—SWE). B: Bacterial counting (log CFU/mL)-plating method. Methanol as a control Extract (ethanol–water). Antibacterial susceptibility—Kirby Bauer disk diffusion method and MIC determination.	Over 0.6 mg/mL of SWE (110 °C) extract exerted a bactericidal effect against *B. cereus* KCCM 40935 (G+, spore-forming, rod-typed, heat-resistant pathogenic) and 1.2 mg/mL of SWE (160 °C) extract exerted a bacteriostatic effect during culturing.	D	[17]
		A: Extract (using subcritical water—SWE). B: Bacterial counting (log CFU/mL)-plating method. Quercetin as a control.	Extract (using subcritical water—SWE). Bacterial counting (log CFU/mL)-plating method. Quercetin as a control.	D	[18]
Carrot (Daucus carota):					
Carrot preparations: orange carrot root; orange carrot pomace; purple carrot root, purple carrot pomace.	Anthocyanins, phenolic acids, carotenes, fibre	A: The fresh material was immediately ground (root) and then dried (root and pomace) in a vacuum dryer at 45 °C for 12 h, grounded and passed through a 0.5-mm mesh sieve to obtain a fine dietary component. B: In vivo experiment with rats. Experimental groups included animals obtained diet with 10% dried preparations; SCFA-GC; bacterial enzymes measured by the rate of release of p-nitrophenol or o-nitrophenol from the respective nitrophenyl glucosides.	In comparison to the control group (standard diet), a significant increase in bacterial caecal *α*- and *β*-glucosidase, *α*- and *β*-galactosidase, and *β*-glucuronidase activity was noted in all four carrot groups. In comparison to the control group, a significant increase in all caecal SCFA concentrations followed all four carrot dietary treatments.	D, ID	[19]
Carrot pomace (with pressed pulp and skin residues		A: Extract (water). B: MIC determination.	Carrot pomace extracts exhibited inhibitory activity against two methicillin-resistant *S. aureus* G+ strains (MRSA 1 and 3) and *Enterococci* 44 (HLAR-VRE). The effectiveness of phenolic compounds was only against G+ bacteria. There was no effect on G− ones: *Pseudomonas* 30, *Klebsiella*, *E. coli* ESBL 365, *E. coli* 2280).	D	[20]
Soybean (Glycine max):					
Tofu whey wastewater	Lactic acid (2.67 g/L); acetic acid (1.87 g/L); total viable count of bacteria (3 × 10^9^ CFU/mL): *Lactobacillus*, *Acetobacter*, *Burkholderiaceae*, *Actinobacteria*, other.	A: Fresh tofu whey wastewater was collected from three township tofu processing factories and was naturally fermented for 5 days at 25 °C. B: Microbiota analysis—PCR based on 16S rDNA. Chickens were infected with Salmonella enteritidis. The chickens obtained drinking water containing tofu whey wastewater for 7 days.	Reduction in the colonisation and excretion of *Salmonella enteritidis* in chickens. Tofu whey wastewater supplementation significantly upregulated the relative abundance of *Lactobacillus* and *Burkholderia* in control and *Salmonella enteritidis*-infected chickens.	D	[21]
Okara-by-product generated from soymilk or tofu production	Basic characteristic was identified (proteins, carbohydrates, lipids dietary fibre, isoflavone).	A: Different dosages of okara (7.5% and 15%) B: In vivo experiment with mice (26 weeks of feeding; n = 11–15 per group). Cecal microbial analysis was conducted using the terminal restriction fragment length polymorphism (T-RFLP) method and was subjected to 16S rDNA. Control—mice with a standard diet.	The relative abundance of *Clostridiales*, *Bacteriodales*, and *Ciriobacteriales* was significantly increased by 15% okara diet supplementation compared to control mice. *Lactobacillus*, *Erysipelotrichaceae*, *Parasutterella*, were significantly decreased in the 15% okara group compared to the control group.	D	[22]
Broccoli (Brassica oleracea):					
stalks; (total fibre fraction, and insoluble fibre fraction, freeze-dried broccoli stalks	Glucosinolates, polyphenols (sinapic acid and chlorogenic acid derivatives), dietary fibre, carbohydrates.	A: Extract (ethanol and water). The samples were digested by a simulated gastrointestinal model. B: SCFA were fermented in in vitro human faecal fermentation model and analysed by GCL chromatographic analysis.	SCFA production increased during the fermentation of both extracts by the microbiota. Insoluble fibre fraction extracted from fresh broccoli stalks exhibited a greater prebiotic effect than freeze-dried broccoli stalks, leading to a higher content of total SCFAs with significant differences in the production of acetate, butyrate and other minor SCFAs (isobutyrate, isovalerate, valerate, isocaproate, caproate and heptanoate).	D	[23]
stalks; leaves, stems and inflorescence; dietary-fibre-rich fractions isolates	Crude protein, fat, uronic acid (pectin), neutral sugars (glucose, xylose, arabinose, fructose, rhamnose, manose, galactose).	A: Total dietary fibre extraction—alcohol insoluble residues method. Modifications: supercritical fluid and enzyme treatments. B: The growth capacity—plating on MRS broth. Evaluation by comparing the percentage of growth in each extract with the positive control (glucose). SCFAs produced in the presence of fibre extracts were determined by a gas chromatograph with a split/split-less injector and a flame ionization detector.	The effect of soluble DF from different parts of the broccoli plant and with different modifications (supercritical fluid and enzyme treatments) on the growth capacity of LAB (*L. sakei, L. brevis, L. plantarum, L. casei* and *Enterococcus faecium*). *L. sakei*, *Enterococcus faecium* and *L. *casei** showed significant differences in growth rates in the presence of the different extracts. DF extracts from leaf and stem samples showed the highest growth values, while DF extracts from the inflorescences showed the lowest values. In contrast, no significant differences were observed in the growth of *L. plantarum* and*L. brevis* in the presence of the different extracts. Treatment with enzymes improved especially the growth of LAB and the production of all the SCFA (acetic, propionic, butyric, isovaleric, isobutyric, isocaproic, caproic, valeric).	D	[24]
Sugar beet (Beta vulgaris):					
pulp	Pectin oligosaccharides, monosaccharides representative for the pectin, i.e., galacturonic acid as acidic sugar, and arabinose, galactose, rhamnose	A: Enzymatic (cellulase) and nitric acid extracted, hydrolysed and fractionated. B: The growth of inoculated cells—impedance microbiology (time to detection coincides with the reaching of a cell concentration of about 10^6^–10^7^ cells/mL). Probiotic effect–compared to growth of the species in not supplemented MRS and TSB broths.	Pectin oligosaccharide compounds promoted the growth of LAB. Not all the fractions worked with the same efficiency stimulating LAB, and that pectin oligosaccharides containing a low degree of polymerization arabinans, and little or no free galacturonic acid (and possibly no nitrates), obtained by enzymatic extraction, were the most efficient. No fraction was able to stimulate pathogenic *E. coli* strains (K88 and K89).	D	[25]
Cabbage:					
Chinese cabbage (Brassica rapa)	Fiber	A: Hydrolyzates; Alcohol-Soluble Fiber (ASF). Two commercial enzymes, Shearzyme Plus and Viscozyme L, were used in the study. B: Growth of intestinal microbiota (*L. plantarum* ATCC 8014, *L. casei* ATCC 393*, L. delbrueckii subsp. bulgaricus* ATCC 11842, and *L. acidophilus* ATCC 832) were investigated using broth microdilution method and plating on MRS.	Alcohol-soluble dietary fibres were found to promote the growth of LAB. ASF had a significantly greater effect on the growth of all LABs except for *L. casei* than the control. Especially, *L. plantarum* and *L. delbrueckii* grew best in the ASF produced by Shearzyme.	D	[26]
Cabbage (Brassica oleracea var. Acephala) stalk flour	Sulfur compounds, such as dimethyl trisulfide, and terpenic compounds, such as phytol and its derivatives, furfural	A: Extracts (aqueous, methanolic, ethanolic) and essential oil. B: Diffusion antimicrobial susceptibility–plating on Mueller–Hinton agar. DIZ (in millimetres) formed around the discs containing the extracts. Negative control-sterilized disc. Positive control—amoxicillin and potassium clavulanate.	The cabbage stalk flour essential oil appeared to be active as an antimicrobial agent against *Salmonella sp.* (G−)*, B. cereus* (G+), *S. aureus* (G+), and *E. coli* (G−). The methanolic extract was active against *E. coli* and the aqueous extract against *S. aureus*, and *E. coli. Listeria monocytogenes* (G+) were not sensitive to all treatments with cabbage stalk flour extracts and essential oil.	D	[27]
Chicory (Cichorium intybus)					
root, peel, leaves, seeds	Polyphenolics	A: Drying, 75% ethanol extraction B: In vivo experiment with rats; Diet supplemented with (a) 10% of root extract (PL); (b) 6.5% of peel extract (PM); (c) 8% of peel extract and 0.8% of seed extract (PH); (d) 2.5% of leaf extract with 0.106% of total phenolics (PMc); (e) control—without phenolics. SCFA-GC; faecal bacterial enzymes—the rate of release of p-nitrophenol or o-nitrophenol from the respective nitrophenyl glucosides. Bacterial enzymes were measured by the rate of p- or o-nitrophenol, according to the Juśkiewicz et al. [28] method.	Diet supplementation with the preparations examined did not result in any significant differences in *β*-glucuronidase activity on day 7, while on day 14, its activity in the PMc group was significantly higher than in the PM and PH diets. The *β*-glucosidase on day 21 was significantly lower in the PM and PMc vs. C. After 3 and 4 weeks, *β*-glucuronidase activity was highest in the C and differed significantly from the PM and PH groups. After 4 weeks, *β*-glucosidase activity in the C and PL groups was significantly higher than in the other groups.	D	[29]
roots flour	Fibre, phenoliccompounds	A: Flour was obtained by drying comminuted roots and ground. B: A 36-day experiment carried out on 54-day-old rabbits fed a diet with the chicory flour at 0, 25 and 50 g/kg. Control—commercial and antibiotic-free diet.	Supplementation of a diet with a chicory flour preparation (both levels) resulted in the lowering of the bacterial enzyme activity in the caecum and colon.	D	[30]
root	Fibre, phenoliccompounds, such as caffeoylquinic acids (CQAs) more specifically mono- and di-CQAs isomers	A: Meal from chicory roots obtained from industrial processing and commercial preparation of FOS produced via the enzymatic hydrolysis of chicory inulin. B: In vivo experiment—Wistar rats with a model of TNBS-induced colitis. Control—diets with dietary cellulose.	Both chicory preparations significantly reduced the pH value of colonic digesta and favourably lowered the caecal activity of bacterial glucuronidase as well as the caecal concentration of putrefactive SCFA in comparison to the control TNBS rats.	D, ID	[31]
Potato (Solanum tuberosum):					
the by-product from potato starch (including the potato residue)	Fibre	A: Freeze-dried, lyophilized, crushed, sieved; ethanol extraction, enzymatic modification of insoluble DF by cellulase and xylanase hydrolysis B: C57BL/6 mice intragastrically fed with 20 mL/(kg d) of low (0.25 mg/(g d), medium (0.50 mg/(g d)), and high dose (1.00 mg/(g d)) of unmodified, or modified potato residue DF. Control-fed with water. SCFA-GC.	Potato residue DF regulated the SCFA production. Unmodified and enzymatic-modified DF extracted from potato residue could promote the production of acetic, n-butyric, isobutyric, valeric, and isovaleric acids while inhibiting the production of propionic acid. DF significantly improved the number and diversity of intestinal microbiota of mice, in particular, the increased ratio of *Bacteroidetes* to *Firmicutes*. Cellulase/xylanase improved regulating effects of dietary fibre on gut microbiota.	D	[32]

The different letters in the Methods column denote A: the form of the by-product/isolation of different components; B: experimental and/or analytical method. Mechanisms of action affecting the microbiota or bacterial environment: DM—direct; ID—indirect. LAB—lactic acid bacteria; *L.*—*Lactobacillus*; *L. casei*—new taxonomic name *Lacticaseibacillus casei*; *L. brevis*—new taxonomic name *Levilactobacillus brevis*; *L. plantarum—*new taxonomic name *Lactiplantibacillus plantarum; E.*—*Escherichia; S.—Staphylococcus;* DF—dietary fibre; FOS—fructooligosaccharides; MRS—Mann Rogosa Sharpe broth; TSB—Tryptone Soya Broth; G+—Gram-positive; G−—Gram-negative; SCFA—short-chain fatty acids; MIC—minimum inhibitory concentration; MBC—minimum bactericidal concentration; MFC—minimum fungicidal concentration; DIZ—diameters of the inhibition zone; BGL—*β*-glucosidase; GUS—*β*-glucuronidase; GC—gas chromatography; TNBS—trinitrobenzenesulfonic acid.

## Data Availability

No new data were created or analyzed in this study. Data sharing is not applicable to this article.

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
