# Peer review of "Side Streams of Vegetable Processing and Its Bioactive Compounds Support Microbiota, Intestine Milieu, and Immune System"

_molecules, 2023, doi:10.3390/molecules28114340_

Round 1
Reviewer 1 Report
The main question addressed by the research is What The importance of side streams as a source of healthy substances with the potential to improve health. especially their effects on the gut environment, immune system, and macrobiotic?
I consider the topic relevant in the field and the address does a specific gap in the field.
- It is more comprehensive than other publications as it includes several aspects at the same article which included Bioactive compounds derived from vegetable wastes have the potential to be employed as excipients in pharmaceuticals and as food additives and performed a narrative review of the current literature in this points.
- It should only take into account the separation between the theoretical part and the research results.
I have no comments on the tables and figures except it would have been better if he added scientific reasons for the results in the table 1 (The effect of side streams of vegetable processing on the micro biota.)
need to improved.
The main question addressed by the research is What The importance of side streams as a source of healthy substances with the potential to improve health. especially their effects on the gut environment, immune system, and macrobiotic?
I consider the topic relevant in the field and the address does a specific gap in the field.
- It is more comprehensive than other publications as it includes several aspects at the same article which included Bioactive compounds derived from vegetable wastes have the potential to be employed as excipients in pharmaceuticals and as food additives and performed a narrative review of the current literature in this points.
- It should only take into account the separation between the theoretical part and the research results.
I have no comments on the tables and figures except it would have been better if he added scientific reasons for the results in the table 1 (The effect of side streams of vegetable processing on the micro biota.)
Author Response
Response to Reviewer 1 Comments
Review: The main question addressed by the research is What The importance of side streams as a source of healthy substances with the potential to improve health. especially their effects on the gut environment, immune system, and macrobiotic? I consider the topic relevant in the field and the address does a specific gap in the field.
Point 1: It is more comprehensive than other publications as it includes several aspects of the same article which included Bioactive compounds derived from vegetable wastes that have the potential to be employed as excipients in pharmaceuticals and as food additives and performed a narrative review of the current literature in this points. It should only take into account the separation between the theoretical part and the research results.
Response 1: We thank the Reviewer for the comment. We agree that the separation of the theoretical part and the research part is often used approach in reviews and could have been implemented in this article. However, the authors choose the approach to focus the attention of the reader on the particular topic and discuss in the theoretical part some issues next to the research results (always supported by relevant literature). The authors kindly request to accept this approach.
Point 2: I have no comments on the tables and figures except it would have been better if he added scientific reasons for the results in Table 1 (The effect of side streams of vegetable processing on the microbiota.)
Response 2: We thank the Reviewer for the valuable comment. According to scientific reasons for the results concerning Table 1, the authors added a description of the direct and indirect mechanisms of action of bioactive compounds and how they can induce the functional and compositional change of microbiota. In Table 1 the column with the mechanism category (direct or indirect) has been included.
“The bioactive compounds can modulate the microbiota directly or indirectly. The mechanisms of direct interactions could include the modulation of (1) the composition (e.g. through inhibition of nucleic acid synthesis, membrane integrity, and permeability, energy metabolism [14-15, 17-18, 26, 33]); (2) the colonization of bacteria [21]; (3) the release of bacterial metabolites (e.g. SCFA) [16, 19, 23-24, 31-32, 34]; (4) the enzyme activity of gut microbiota (e.g. β-glucosidase, β-glucuronidase activity)[19]. The indirect interactions affecting bacteria could also occur through (1) modulation of the pH in the intestine [16, 31] and (2) the gastrointestinal transit time [35], (3) the synthesis and release of antimicrobial peptides [36].”
The additional references have been included in the Reference section:
[33] Maurya, A.; Prasad, J.; Das, S.; Dwivedy, A.K. Essential Oils and Their Application in Food Safety. Front. Sustain. 2021, 5, doi:10.3389/fsufs.2021.653420.
[34] Feng, W.; Liu, J.; Cheng, H.; Zhang, D.; Tan, Y.; Peng, C. Dietary Compounds in Modulation of Gut Microbiota-Derived Metabolites. Front Nutr. 2022, 9, doi:10.3389/fnut.2022.939571.
[35] Slavin, J. Fiber and Prebiotics: Mechanisms and Health Benefits. Nutrients 2013, 5, 1417–1435, doi:10.3390/nu5041417.
[36] Hou, X.; Li, S.; Luo, Q.; Shen, G.; Wu, H.; Li, M.; Liu, X.; Chen, A.; Ye, M.; Zhang, Z. Discovery and Identification of Antimicrobial Peptides in Sichuan Pepper (Zanthoxylum Bungeanum Maxim) Seeds by Peptidomics and Bioinformatics. Appl. Microbiol. Biotechnol. 2019, 103, 2217–2228, doi:10.1007/s00253-018-09593-y.
Point 3: Comments on the quality of english language - need to improved.
Response 2: Linguistic changes have been made.
Reviewer 2 Report
The subject of the article is very interesting, but for a more complete approach I think that an important addition is needed. It is known that vegetables bring valuable nutrients for the human body and for the intestinal microbiome represented by fibers, antioxidants, vitamins, minerals. It is very important to specify the intake of vitamins and minerals with beneficial effects on the microbiota, and if we consider the seasoning plants here, it is clear that the volatile oils also contain compounds with a significant effect on the microbiota. On the other hand, in vegetables we also have antinutrients that can influence the microbiota.
After a reanalysis of the manuscript, I better specify the aspects proposed to the authors to improve the presentation: completing the discussions with the impact of anti-nutrients from vegetable products (lectins, phytates, oxalates, etc.) on the human body and especially on the permeability of the intestinal membrane and the need to mitigate the negative effects, because there is a fashionable trend to stimulate the consumption of vegetable products but the most it is often not specified that this category of food products is also the richest in anti-nutrients with sometimes particularly serious negative effects on the body, especially in people who adopt the row vegan diet in the long term, many allergic, intolerant disorders appear or even autoimmune diseases, because often the thermal processing and especially with steam under pressure without exaggeration in this way to protect the valuable nutrients. I reconsider the recommendation as a minor revision.
Author Response
Dear Reviewer,
Please see the attachment with Authors' responses to Reviewer's comments.
Best regards,
Joanna Fotschki

Reviewer 3 Report
I would like you to correct on page 7 line 154 in the case of examples of phenolic acids, that in fact they are carboxylic acids without presenting the hydroxy functional group, in order to be phenolic acids.
Author Response

(The authors gave the same response as above.)

Round 2
Reviewer 2 Report
I accept the publication in a revised form, but pay attention to the mistakes in the text: there are extra punctuation marks (dots), line 382 for example.
Reviewer 3 Report
I would suggest that in table 1 you fill in the mechanism of action, instead of the mechanism
I would complete in the same table 1 instead of D, DM, as you mentioned in the legend
I would suggest that on line 225, you put the bacterial strain instead of the bacteria strain (Figure 1)